# Insight into the Interaction of Malondialdehyde with Rabbit Meat Myofibrillar Protein: Fluorescence Quenching and Protein Oxidation

**DOI:** 10.3390/foods12102044

**Published:** 2023-05-18

**Authors:** Xiaosi Chen, Zhifei He, Zefu Wang, Hongjun Li

**Affiliations:** 1College of Food Science, Southwest University, No.2 Tiansheng Road, Beibei District, Chongqing 400715, China; 2Chongqing Engineering Research Center of Regional Food, No.2 Tiansheng Road, Beibei District, Chongqing 400715, China; 3Chongqing Key Laboratory of Speciality Food Co-Built by Sichuan and Chongqing, No.2 Tiansheng Road, Beibei District, Chongqing 400715, China; 4College of Food Science and Technology, Guangdong Ocean University, Zhanjiang 524088, China

**Keywords:** mediated oxidation, structural change, addition reaction, dynamic quenching, first-order kinetics

## Abstract

This research explored the effects of oxidative modification caused by different malondialdehyde (MDA) concentrations on rabbit meat myofibrillar protein (MP) structural characteristics and the interactions between MDA and MP. The fluorescence intensity of MDA–MP adducts, and surface hydrophobicity increased, whereas the intrinsic fluorescence intensity and free-amine content of MPs decreased as MDA concentration and incubation time increased. The carbonyl content was 2.06 nmol/mg for native MPs, while the carbonyl contents increased to 5.17, 5.57, 7.01, 11.37, 13.78, and 23.24 nmol/mg for MP treated with 0.25 to 8 mM MDA, respectively. When the MP was treated with 0.25 mM MDA, the sulfhydryl content and the α-helix content decreased to 43.78 nmol/mg and 38.46%, while when MDA concentration increased to 8 mM, the contents for sulfhydryl and α-helix decreased to 25.70 nmol/mg and 15.32%. Furthermore, the denaturation temperature and ΔH decreased with the increase in MDA concentration, and the peaks disappeared when the MDA concentration reached 8 mM. Those results indicate MDA modification resulted in structural destruction, thermal stability reduction, and protein aggregation. Besides, the first-order kinetics and Stern–Volmer equation fitting results imply that the quenching mechanism of MP by MDA may be mainly driven by dynamic quenching.

## 1. Introduction

Rabbit meat provides excellent nutritive and good properties, making it popular among consumers [1]. However, rabbit meat is rich in protein and unsaturated fatty acid, which means it is prone to spoilage due to microbial reproduction and oxidation of lipids and proteins [2]. Lipids and proteins coexist in meat products; lipid oxidation is more naturally occurring than protein oxidation. Malondialdehyde (MDA), the major aldehyde originating from lipid peroxidation, is produced during commercial cryopreservation [3].

Protein oxidation can influence the physicochemical properties of meat and thus affect the quality of meat products [4,5,6]. Myofibrillar protein (MPs), the main component of meat proteins, accounts for 55–60% of meat proteins [7]. Therefore, the textural properties of meat and meat products are closely related to the quality of MPs [8]. Thus, to maintain good texture properties of the meat product, the regularity of protein oxidation mediated by MDA needs further investigation, especially the binding capability of MPs with MDA. Protein oxidation can be induced by MDA; the increase in carbonyl content and decrease in sulfhydryl content are dose-dependent on the content of MDA [9,10]—However, limited studies centered on the non-covalent and covalent interactions between MDA and MPs. Instead, most research focused on the oxidant effects of MDA on MPs. Studies also rarely reported the binding mode and the interaction rate between MDA and MPs.

The effects of different concentrations of MDA added in the rabbit meat MP system on MP oxidation and structural characteristics were investigated in the current study. Furthermore, endogenous fluorescence spectroscopy was used to explore the fluorescence quenching effect of different concentrations of MDA on rabbit meat MP. Stern–Volmer equation and first-order kinetics were used to determine the function type and the interaction mechanism.

## 2. Materials and Methods

### 2.1. Materials

Two hundred and fifty-two male Ira rabbit carcasses (weight: 2.3–2.5 kg/live rabbit, age: 75 days) were purchased from Chongqing A Xing Ji Food Co., Ltd., (Chongqing, China). The rabbits were electro-stunned before slaughter. After being slaughtered by standard commercial procedures, the carcasses were aged at 4 °C for 24 h to dissipate rigor mortis. Five hundred and four LTL (Longissimus thoracis et lumborum) samples were dissected, samples were placed in an approximately 6 °C chilled box, and immediately transported to the laboratory. Then, removed the connective tissue. All samples were randomly divided into seven groups; hence, every group had 72 LTL samples, so every independent experimental trial had 24 samples. The LTL samples were placed in an ultra-low temperature refrigerator (−80 °C). The freezing LTL samples were thawed for 12 h in a 4 °C incubator, followed by mincing of the meat by using a blender. All chemicals were of analytical grade or higher.

### 2.2. Extraction of MPs

MPs were extracted from rabbit LTL following a method reported previously [9]. Subsequently, the MPs were suspended in 20 mM phosphate buffer (pH 6.0) containing 0.6 M NaCl, after which the suspension was filtered by four layers of cheesecloth. Finally, the protein concentration was adjusted to 20 mg/mL by the Biuret method.

### 2.3. Preparation of MDA Solution

The MDA solutions were prepared following a previously reported method [11]. The absorbance of the MDA solution at 267 nm was measured with an ultraviolet (UV) spectrophotometer, and a molar extinction coefficient (31,500 M^−1^ cm^−1^) was used to calculate the MDA concentration.

### 2.4. Incubation of MP with MDA

A total of 20 mg/mL MP suspension (containing 0.5 mg/mL NaN_3_) was prepared with phosphate buffer (20 mM, pH 6.0). Then, the seven MPs suspensions groups were added with different contents of MDA to achieve the final concentrations of MDA in protein solutions: 0, 0.25, 0.5, 1, 2, 4, and 8 mM/L. Every group had nine parts, which means each MDA concentration treatment has three independent parallel parts at one temperature. The resulting solutions were then incubated at 4 °C, 14 °C, and 24 °C and were shaken well away from the light. The solutions were centrifuged at 4 °C to obtain the sediment. The sediment was washed thrice with 20 mM phosphate buffer and centrifuged under the same condition to remove the residue MDA. The samples incubated in the 4 °C groups were analyzed at 1, 2, 4, 8, 12, and 24 h, whereas those incubated in the 14 °C and 24 °C groups were analyzed at 24 h.

### 2.5. Measurement of Carbonyl Content of MP

The carbonyl content of the MPs was used in the 2,4-dinitrophenylhydrazine (DNPH) colorimetric method [12].

### 2.6. Measurement of Sulfhydryl Content of MP

The indicator 5,5′-dithiobis (2-nitrobenzoic acid) (DTNB) was used to measure the total sulfhydryl content in the treated MPs [13].

### 2.7. Measurement of Free-Amine Content

The ophthalaldehyde (OPA) method was used to determine the free-amine content of MPs based on previous studies with slight modifications [14]. The absorbance of the mixture at 340 nm was measured using a standard curve constructed from glycine with a UV spectrophotometer (UV1770, Shimadzu, Kyoto, Japan).

### 2.8. Secondary Structure of MP

The Secondary structure of MPs was measured by the method of Jia et al. [15]. The concentration of MPs was diluted to 20 mg/mL protein by adding 20 mM phosphate buffer containing 0.6 M NaCl with pH 6.0. The Raman spectrometer (ThermoFisher Scientific, DXR2, Waltham, MA, USA), with a wavelength from 3600 cm^−1^ to 400 cm^−1^, was used to conduct the test. The information on secondary structural components can be studied from the amide region I (1700–1600 cm^−1^). The collected spectral data were analyzed with Peak Fit version 4.12 software.

### 2.9. The Intrinsic Tryptophan Fluorescence of MP

The tryptophan fluorescence spectrum of the MP samples was obtained based on a previously reported method [16]. The concentration of the MP sample was adjusted to 0.5 mg/mL and recorded from 300 nm to 400 nm when the excitation wavelength was 280 nm.

### 2.10. Determination of UV Absorption Spectra of MPs

The UV spectrum of MPs was recorded from 230 nm to 320 nm using a spectrophotometer (Shimadzu, UV-16001, Kyoto, Japan). 

### 2.11. Determination of the Fluorescence Intensity of MDA–MP Adducts

The fluorescence intensity of MDA–MP adducts were determined based on the method reported before but with slight modifications [17]. The MPs sample was adjusted to 0.5 mg/mL by 20 mM phosphate buffer. The wavelength of the recorded spectrum was between 400 and 600 nm, with the excitation wavelength at 390 nm.

### 2.12. Surface Hydrophobicity

The surface hydrophobicity of MP was measured by referring to a method reported before but with slight modifications [8]. The absorbance at 595 nm was used to determine the amount of bound BPB based on the following formula:(1)$$BPB\left(\mu g\right)=200\times \frac{A_{Control}-A_{Sample}}{A_{Control}}$$

### 2.13. Differential Scanning Calorimetry (DSC)

Aluminum pans (TAQ 10, 6.65 × 1.7 mm) were used to seal the MPs hermetically, and every pan was filled with 10 mg sample. Then, the pans were heated from 30 °C to 110 °C at a rate of 10 °C/min, and the empty pan was considered blank. The enthalpy change (ΔH) and peak denaturation temperatures (Td) were recorded by DSC (Waters, DSC-25, Milford, MA, USA).

### 2.14. Measurement of Electrophoresis of MP

The degree of protein cross-linking was determined using SDS-polyacrylamide gel electrophoresis (SDS-PAGE) [15]. Briefly, 5% and 10% acrylamide were used to prepare the stacking and separating gels. Then, the sample buffer was mixed with a certain concentration of MPs. Several groups were added with 10% β-ME, whereas the others were not. Then, 10 μL of each sample was loaded into a well in the gel after the treatment with boiling water for 5 min.

### 2.15. Kinetic Modeling

First-order kinetics fit the data to explore the MDA and MP adduct reactions further. In most cases, most reactions conform to first-order kinetics [18,19,20]. Thus, the addition reaction of MDA with MP can be described as a first-order reaction, as shown in Equations (2) and (3). The fractional conversion model is a widely used kinetic model that measures the extent of a reaction to evaluate the corresponding changes; this model is defined as Equations (4) and (5) [21,22].
(2)$$\frac{d_{C}}{d_{t}}=-k\times C$$
(3)$$ln\frac{C_{t}}{C_{0}}=C_{0}-k\times t$$
(4)$$f=\frac{C_{0}-C_{t}}{C_{0}-C_{\infty }}$$
(5)$$\ln\left(1-f\right)=ln\frac{C_{0}-C_{t}}{C_{0}-C_{\infty }}=-k\times t$$
where *f* is the MDA–MP adducts, *C*_0_ is the initial fluorescence intensity value, *C_t_* is the initial fluorescence intensity value at time *t*, *C_∞_* is the initial fluorescence intensity value at time *t* with prolonged heating, and *k* is the reaction rate constant. The first-order kinetics models were established by a partial least squares regression algorithm and were verified by leave-one-out cross-validation (IBM, SPSS 20.0, Chicago, IL, USA).

### 2.16. Statistical Analysis

All determinations for each assay consisted of three independent experimental trials, and each replicate was conducted in triplicate. A one-way analysis of variance (SPSS 20.0, USA) was used to analyse the effects of MDA on MP oxidation. Duncan’s multiple range test was used to determine the difference between means, and the significance was defined as α = 0.05.

## 3. Results and Discussion

### 3.1. Changes in Amino-Acid Side Chains

#### 3.1.1. Carbonyl Content

Figure 1A shows the carbonyl content in native (0 mM MDA) and MDA-modified MP. The initial carbonyl content of native MP was 2.06 nmol/mg. As the incubation time increased, the carbonyl contents in native MP increased. Still, not significantly (*p* > 0.05), and this finding may be due to the free radicals generated by residual oxygen in carbonylate protein [23]. The carbonyl contents of MP incubated with MDA increased significantly *(p* < 0.05) with the increase in incubation time. Meanwhile, MDA addition dose-dependently increased the carbonyl content. It is quite evident that the carbonyl content was significantly promoted (*p* < 0.05) in treated groups (0.25, 0.5, 1-, 2-, 4-, and 8-mM MDA addition) and increased by 138%, 148%, 187%, 303%, 368% and 620%, respectively, compared to the control after 24 h incubation. These results indicate that amine groups on side chains are readily modified and converted into carbonyls under MDA-induced oxidative stress [24,25]. In addition, the carbonyl content increased linearly (*R*^2^ = 0.982) with the increased MDA concentration after 24 h of incubation. As the incubation time of MPs treated with the same concentration of MDA was prolonged, the increase rate of carbonyl groups decreased gradually. This result may be attributed to the protein–protein cross-links under MDA-induced oxidative stress.

#### 3.1.2. Sulfhydryl Content

The sulfhydryl content in MP decreased as the MDA concentrations and incubation time increased (Figure 1B). The sulfhydryl content of native MP was 60.37 nmol/mg MP, which was similar to the result of previous studies [2]. The decrease in sulfhydryl content depended on the time of storage. After 12 h incubation, the sulfhydryl content of the control group was 58.60 nmol/mg protein, while it was 43.78, 43.57, 38.01, 36.28, 35.93 and 25.70 nmol/mg protein in MP with 0.25, 0.5, 1-, 2-, 4-, and 8-mM MDA addition, respectively. Compared with native MP, about 30%, 32%, 35%,40%, and 57% sulfhydryl contents were lost after 0.25, 0.5, 1-, 2-, 4-, and 8-mM MDA addition after incubation for 24 h, respectively; this condition indicates that MDA-induced oxidative stress can lead to a decrease in sulfhydryl content [8]. As we know, the sulfur-containing amino acids were located in hydrophobic inner areas when the MPs were attacked by MDA, which led to the sulfur-containing amino acids being exposed to the surface, which may lead to structural changes, such as aggregation, polymerization, and degradation [7,11].

#### 3.1.3. Free-Amine Content

Two reactive aldehyde groups in MDA formed a Schiff base complex with amino-acid side chains, especially the ε-amino groups of lysine residues [11]. Figure 1C shows the effects of MDA concentration and incubation time on the free-amine content of MP. The amines of MP steadily decreased as MDA concentration and incubation time increased. The free-amine content of native MP was 102.55 nmol/mg and was reduced by 10% after treatment for 24 h at 4 °C, so maybe the oxygen on top of the bottle reacted with protein. However, in 0.25 mM MDA, the free-amine content decreased to 61.20 nmol/mg MP. It was observed that the free-amine content decreased as the MDA concentration increased, and the free-amine content decreased by 50% when the MDA concentration reached 1 mM. After incubation for 24 h, the free-amine content of MP treated with 8 mM MDA was reduced by 75%, and that of MP treated with low doses of MDA (0.25 mM) was reduced by 41%. Thus, the amine groups on side chains are readily modified and converted into carbonyls under MDA-induced oxidative stress [7]. These results indicated that MDA interacted with MP showed dose and time-dependently and agreed with the analysis in carbonyl and sulfhydryl.

### 3.2. Surface Hydrophobicity

The extent of hydrophobic amino acid residues distributed on protein surfaces can be evaluated by surface hydrophobicity [13]. The initially occluded hydrophobic amino acids were exposed by oxidation to the polar surface. As shown in Figure 1D, the surface hydrophobicity of MP gradually increased as the MDA concentration and incubation time increased. The surface hydrophobicity reached a maximum when MP was incubated with 8 mM for 24 h, more than twice that of the native MP (52.24 μg BPB). Besides, when the incubated time was less than 12 h, the surface hydrophobicity of MP increased almost linearly for all concentration MDA addition. In comparison, when the incubated time exceeded 12 h, the increase rate decreased. This phenomenon is attributed to the exposure of hydrophobic groups, which reflects the structural changes in MP under MDA-induced oxidative stress. Hydrophobic residues, especially nonpolar aromatic amino acids, were mainly obscured and folded inside the protein. The surface hydrophobicity of MP treated with MDA gradually increased, suggesting that MDA promoted the unfolding of the protein, which accords with the result of previous studies [8,10].

### 3.3. Fluorescence Intensity of MDA–MP Adducts

Figure 2A shows the fluorescence intensity of MDA–MP adducts in MP treated with MDA at 4 °C for 24 h at different concentrations. MDA–MP adducts had a strong fluorescence emission peak at around 446 nm, whereas no fluorescence intensity was detected in native MP. As for the samples treated with MDA, the fluorescence intensity of the MDA–MP adduct increased with the increase in MDA concentration and incubation time. This result indicates that the content of the MDA–MP adduct increased with the increase in MDA concentration [8].

The fractional first-order kinetics described well (*R*^2^ from 0.89 to 0.96) the changes in the fluorescence intensity of MDA–MP adducts over time (Table 1). Further linear fitting was performed for the MDA concentration and reaction rate. An excellent linear fitting was achieved (*R*^2^ = 0.99) below the 4 mM MDA, whereas above 4 mM MDA, a desirable linear fitting was achieved (*R*^2^ = 0.95). However, it was not better than that obtained at low concentrations (Figure 2B). Thus, at a low concentration of MDA (below 4 mM MDA), MDA was directly added to the free amine of the side chain.

In contrast, when its concentration was higher than 4 mM, MDA occupied sites that can bind directly and then changed the structure of MPs, which led to MPs exposing more binding sites. This explains the linear fitting effect of high-concentration MDA is lower than that of low-concentration. The results are consistent with those of previous research [26,27].

### 3.4. Changes in Secondary Structure

Raman spectra were obtained to investigate further the secondary structural changes induced by MDA. Figure 3A shows the changes in the contents of α-helix, β-sheet, β-turn, and the random coil of the MP treated with different MDA concentrations. The proportion of α-helix decreased as the MDA concentrations increased, whereas the random coil content of MP exhibited the opposite trend. The α-helix contents were 39.77%, 38.47%, 37.52%, 31.39%, 20.90%, 16.94%, and 15.32%, respectively, for the MP incubated with 0, 0.25, 0.5, 1-, 2-, 4-, and 8-mM MDA. While the random coil contents were 16.47%, 19.36%, 17.69%, 27.31%, 31.65%, 32.39%, and 33.40%. Compared with the native MP (0 mM), the MP added with 8 mM MDA caused a 24% decrease in α- helix and a 15% increase in the random coil. In addition, it was observed that when the MDA concentration low than 1 mM, the reduced amount of α-helix was equal to the increased amount of random coil. This phenomenon occurred during protein oxidation, indicating that MDA induces protein oxidation [28,29]. This result was probably caused by the binding of MDA to MP, which can change the micro-environment caused by unfolding. More binding sites were also exposed due to the MDA attack on MP.

### 3.5. Absorption Spectra of MP in Soret Band

The UV absorption spectra of MP, mainly derived from tyrosine and tryptophan residues, reflect the changes in aromatic amino acids in the side chain [30]. The absorption spectra of MP without MDA revealed a broad absorption band peaking at nearly 275 nm (Figure 3B). In the presence of an increased concentration of MDA, the maximum absorption peak gradually decreased. Thus, alterations in the spectroscopic properties of MP were mainly ascribed to the oxidative modification of tyrosine and tryptophan residues on the protein surfaces, possibly resulting from the formation of the ground-state MP–MDA complex [8]. However, several studies showed that the adducts of MP and MDA are not simple nor linear [21]. Furthermore, absorption spectra without any noticeable spectral shift and broadening of the peak were observed in the presence of MDA, implying that the possible quenching mechanism of the fluorescence of MP by MDA involves a dynamic process. Thus, further studies must be designed to explore the type of fluorescence quenching during the interactions between MP and MDA.

### 3.6. Intrinsic Tryptophan Fluorescence

Figure 4A shows the fluorescence emission peak of rabbit meat MP at around 335 nm. The MPs with added concentrations of 0 mM to 8 mM MDA had an ordinal reduction of fluorescence intensity. Tryptophan residues exposed to a hydrophilic environment and oxidized in oxidized MP resulted in reduced fluorescence [31]. The results suggest that MP was unfolded, leading to a further decrease in fluorescence intensity under the oxidation induced by MDA. This result is consistent with the changes in the indicators analyzed above. Fluorescence data were analyzed with the Stern–Volmer equation to further discern the quenching mechanism between MP and MDA [32].
(6)$$\frac{F_{0}}{F}=1+K_{sv}[Q]$$
where represents *F*_0_ the fluorescence intensity of native MP (add 0 mM MDA). *F* denotes the fluorescence intensity at different MDA concentrations, *K_sv_* is the Stern–Volmer quenching constant, and [*Q*] is the MDA concentration. As shown in Figure 4B, the correlation coefficient (*R*^2^) of the Stern–Volmer equation at different temperatures was higher than 0.96, which means that the equation fitted the data well. The values of *K_sv_* increased with the increase in temperature, which implies that the quenching mechanism of MP by MDA may be mainly driven by dynamic quenching [33]. Meanwhile, in MDA oxidation systems, MDA–MP adducts were formed, which indicates that dynamic and static quenching occurred [32]. The fluorescence quenching parameters *K_sv_* represent the molecular interaction between the fluorescence groups of the quencher. In this study, MDA reduced the endogenous fluorescence of protein tryptophan mainly through dynamic quenching, which means MDA collides with MPs and causes the MPs structure to unfold, leading the MPs to expose more hydrophobic groups. Then MDA is further adduced with the free amine acids. This explained an excellent linear fitting in MDA–MP adducts below the 4 mM. Given this point of view, protein oxidation can be controlled by competitive inhibition of the combination between protein and MDA by other components, such as polyphenols or essential oils.

### 3.7. DSC Analysis

The two peak denaturation temperatures were labeled as Td1 and Td2. As shown in Table 2, the denaturation temperatures of native MP (added with 0 mM MDA) were 59.107 °C and 76.730 °C, corresponding to the denaturation temperature of myosin and actin, respectively [7]. The denaturation temperature and ΔH decreased with the increase in MDA concentration. The results illuminate that the structural stability of MP was reduced by MDA-induced oxidation, and the change in denaturation temperature was associated with the contents of α-helix [34]. When the MDA concentration reached 2 mM, the thermal denaturation peak of myosin disappeared, whereas the thermal denaturation peak of myosin disappeared when the MDA concentration reached 8 mM. This finding suggests that myosin is susceptible to denaturation under MDA-induced oxidation. The results demonstrate that MDA can cause MP unfolding, which accords with the result of other studies [17,23].

### 3.8. Cross-Linking

In this study, SDS-PAGE was carried out to monitor the covalent cross-linking and changes in the molecular weight of MP under different MDA concentrations. Figure 5 illustrates the changes in MP under MDA-induced oxidation in the absence (A) and presence (B) of β-ME. Without β-ME, an intensity band accumulated at the top of the stacking gel when the MDA concentration was lower than 1 mM. The strength of the strip increased when the concentration of MDA increased. When the concentration of MDA was higher than 1 mM, the changes in the intensity band showed the opposite trend. In addition, the changes in the band intensity of the myosin heavy chain (MHC) revealed the same trend as the band accumulated at the top of the stacking gel. The results indicate that a low concentration of MDA (lower than 1 mM) causes MP crosslinking, whereas a high MDA concentration (higher than 1 mM) causes the degradation or digestion of MP. In the presence of β-ME, the accumulated bands at the top of the stacking gel reduced in strength but still existed compared with the MP without β-ME, which implies that other covalent bonds were present in the protein during cross-linking but not in disulfide cross-linking. The MHC bands with reduced strength can be recovered, and the intensity band of MHC decreases with the increased concentration of MDA. These results revealed that oxidative aggregation and degradation of protein occurred simultaneously under the MDA oxidation system, protein oxidative aggregation predominated at low concentrations of MDA. In contrast, protein degradation predominated at high concentrations of MDA. This foundation can be observed in previous research [35].

## 4. Conclusions

This paper investigated the interaction of MDA with rabbit meat MPs and its binding mechanism. The results show that the MDA and MPs adduct reactions conform to first-order kinetics, and MDA modification resulted in structural destruction, thermal stability reduction, and protein aggregation. In addition, Stern–Volmer equation fitting results imply that the quenching mechanism of MPs by MDA may be mainly driven by dynamic quenching. The research can provide new insights into controlling protein oxidation through competitive inhibition of protein binding to MDA by natural polyphenols components.

## Figures and Tables

**Figure 1 foods-12-02044-f001:**
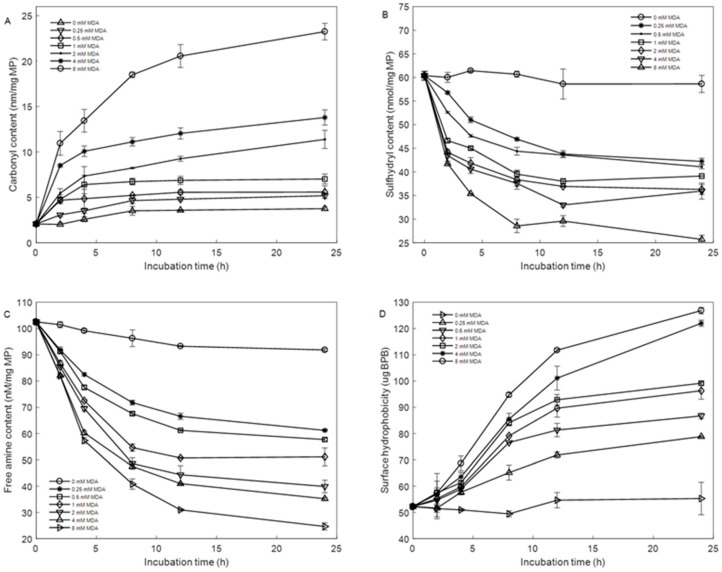
Change in carbonyl content (**A**), sulfhydryl content (**B**), free amine content (**C**), and surface hydrophobicity (**D**) under different MDA concentrations and incubation time at 4 °C. The error bars indicate the standard error obtained from the three analyses.

**Figure 2 foods-12-02044-f002:**
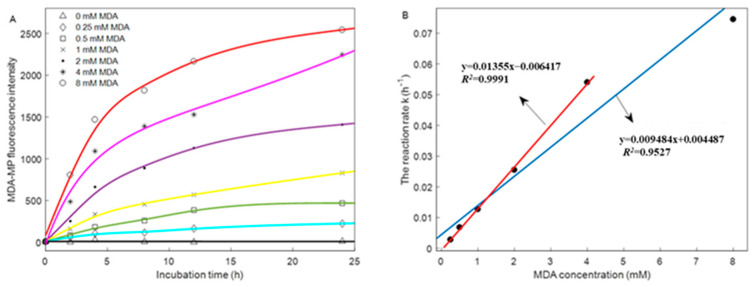
(**A**) Change in the fluorescence intensity of the MDA–MP adducts under different MDA concentrations and incubation time at 4 °C, (**B**) The linear fitting of MDA concentration with the reaction rate constant forming MDA–MP adducts.

**Figure 3 foods-12-02044-f003:**
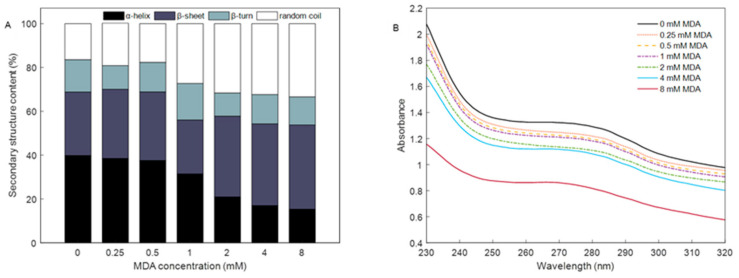
Change in the secondary structure content (**A**) and UV absorption spectra (**B**) of MP induced by different MDA concentrations under 4 °C for 24 h.

**Figure 4 foods-12-02044-f004:**
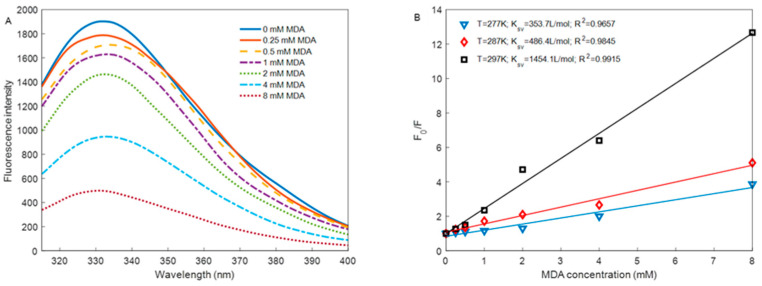
Change in the intrinsic tryptophan fluorescence of MP induced by different MDA concentrations under 4 °C for 24 h (**A**), The Stern–Volmer curves of MP with MDA at various temperatures, and the inset shows the Stern–Volmer quenching constants *K_sv_* at different temperatures (**B**).

**Figure 5 foods-12-02044-f005:**
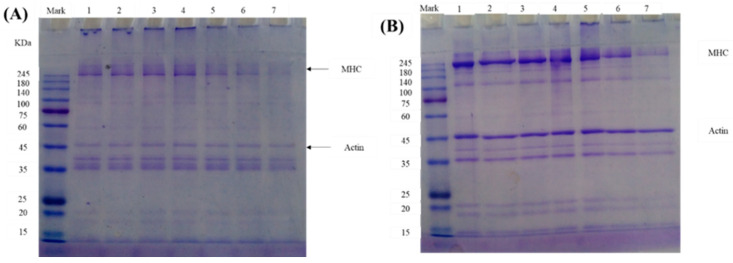
SDS-PAGE of MP incubated with different MDA concentrations under 4 °C for 24 h in the absence of β-ME (**A**) and the presence of β-ME (**B**).

**Table 1 foods-12-02044-t001:** Estimated kinetic parameters of reaction rate for forming the MDA–MP adducts under 4 °C for 24 h at different MDA concentrations.

Kinetic Parameters	Different Concentrations of MDA					
	0.25 mM	0.5 mM	1 mM	2 mM	4 mM	8 mM
k (h^−1^)	0.0028	0.0068	0.0127	0.0255	0.054	0.0745
*R* ^2^	0.9029	0.8959	0.9454	0.9025	0.9672	0.9425

**Table 2 foods-12-02044-t002:** Change in the denaturation temperature (Td) and the total enthalpy (ΔH) of MP induced by different MDA concentrations under 4 °C for 24 h.

MDA Concentration (mM)	Td_1_ (°C)	ΔH_1_ (J/g)	Td_2_ (°C)	ΔH_2_ (J/g)
0	59.107 ± 0.050 ^a^	0.862 ± 0.040 ^a^	76.730 ± 0.333 ^a^	0.430 ± 0.031 ^a^
0.25	58.217 ± 0.343 ^b^	0.323 ± 0.020 ^b^	72.843 ± 0.125 ^b^	0.272 ± 0.021 ^b^
0.5	57.550 ± 0.075 ^c^	0.271 ± 0.012 ^c^	72.817 ± 0.117 ^b^	0.276 ± 0.013 ^b^
1	56.88 ± 0.258 ^d^	0.074 ± 0.036 ^d^	71.767 ± 0.009 ^b^	0.240 ± 0.043 ^b^
2	-	-	69.757 ± 1.094 ^c^	0.172 ± 0.004 ^c^
4	-	-	66.003 ± 0.642 ^d^	0.120 ± 0.016 ^c^
8	-	-	-	-

Values are means ± the standard error (SE). Different letters (a–d) in the same index indicate significant differences between MDA concentrations (α = 0.05).

## Data Availability

The data presented in this study are available on request from the corresponding author.

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
