# Peer review of "Insight into the Interaction of Malondialdehyde with Rabbit Meat Myofibrillar Protein: Fluorescence Quenching and Protein Oxidation"

_foods, 2023, doi:10.3390/foods12102044_

Round 1

Reviewer 1 Report

Following issues/points should be taken into considerations:

LINE 69: Please check the chemical names “MgCl2”  to “MgCl2”

LINE 128:  Please check the units “cm-1” to “cm-1

LINE 209-212: incorrect publication citation “(Wang et al., 2018)- (Feng et al., 2015; Wang et al., 2019; Wu, Zhang & 212Hua, 2009)” please check the manuscript.

LINE 236: Please check the units – “ug” to “μg”

LINE 270: “Ct” to “Ct”

LINE 273: “(R2 from 0.89 to 0.96)” to “R2

LINE 331-334: Check the “Ksv”

 Also; The authors could briefly explain the basis for the increase in sulfhydryl content seen in the control group between 3 and 5 min on average as shown in Figure 1.

Author Response

Referee 1:

Comments to the Author

LINE 69: Please check the chemical names “MgCl2” to “MgCl2

LINE 128:  Please check the units “cm-1” to “cm-1

LINE 209-212: incorrect publication citation “(Wang et al., 2018)- (Feng et al., 2015; Wang et al., 2019; Wu, Zhang & 212Hua, 2009)” please check the manuscript.

LINE 236: Please check the units – “ug” to “μg”

LINE 270: “Ct” to “Ct

LINE 273: “(R2 from 0.89 to 0.96)” to “R2

LINE 331-334: Check the “Ksv”

Reply: Thank you for your suggestions, we have checked the manuscript, and revised the incorrect marks.

Also; The authors could briefly explain the basis for the increase in sulfhydryl content seen in the control group between 3 and 5 min on average as shown in Figure 1.

Reply: Thank you for your suggestions, the sulfhydryl content maintain stable and did not increase the control group, maybe on average it's just fluctuations in the measured data.

Reviewer 2 Report

The publication "Insight into the interaction of malondialdehyde with rabbit meat myofibrillar proteins: fluorescence quenching and protein oxidation" is an original work and very interesting from a substantive point of view.

As a reviewer,  the authors should complete the information on the animal material: the number of animals studied, the conditions of rearing ( including feeding) and pre-slaughter marketing, slaughter conditions, and technique.

The study methods were discussed with great precision and detail.

In addition, the Introduction chapter was treated, in my opinion, too superficially. I propose to make it more detailed.

The description and discussion of the obtained research results were carried out correctly, with reference to correctly selected data from the literature.

Author Response

Reply: Thank you for your suggestions, we have added the information about the materials, Chongqing A Xing Ji Food Co. Ltd. is the largest standard farm of Ira rabbit in China, and the conditions of rearing are standardized. The rabbits were electro-stunned before slaughter, after slaughtered by standard commercial procedures, the carcasses were aged at 4 °C for 24 h to dissipate rigor mortis. We have revised the methods (delete details).

In introduction chapter, we have revised the meaning of the manuscript, and make purpose more detailed, as shown in line 34-39, 45-47.

The discussion section has added appropriate discussion, as shown in line 163-166, 177-180 and 245-251.

Reviewer 3 Report

The manuscript foods-2333542 entitled "Insight into the interaction of malondialdehyde with rabbit meat myofibrillar protein: fluorescence quenching and protein oxidation”

I think that is very interesting paper. Nevertheless, the paper presents some problems and deficiencies.

Abstract

Followed by study objectives, please include material and methods (briefly) in this section prior to the results.

Keywords: Keywords should be different from the title of the manuscript. Please replace the following keywords as they already appeared in the main title

“Malondialdehyde; Rabbit meat myofibrillar protein; Protein oxidation; Fluorescence quenching”

Introduction:

The introduction section does not explain what was done on topic analyzed. Authors must focus that general information and improve this section by including the relevant information about the topic already available in the literature.

Line 34: The authors mentioned low fat content. What does it mean? Is it overall fat or Saturated fatty acids content? I recommend to include figures here showing a brief comparison of total fats with other species (beef, pork, chicken, mutton).

Line 35: Please verify this word “low allergenicity”

Is it correct? I think it should be low atherogenicity index?

Line 36: Why rabbit meat is more prone to microbial spoilage?

This point needs to be explained here with specific references.

Line 40: Please provide a suitable citation here.

Materials & Methods:

This section is by far too long. Thus, detailed description of well-known/published methods is not necessary.

Please include a schematic diagram of the experimental design for better understanding of the readers. In the current form, it is hard to understand about the experimental design. 

Line 59: Time of slaughtering? Because oxidation has a direct relationship with the time.

Line 62: Please specify the method of freezing.

Line 65: Please replace MP with MPs. Its plural, because its many proteins like actin, myosin, troponin, tropomyosin. Also replace was with were here.

2.5 Measurement of carbonyl content of MP

Delete the following paragraph, I think not needed as you already mentioned the reference of the published paper that you are following this method without modification.

“Briefly, acidic conditions were attained to allow MP to react with DNPH. Then, the MP was precipitated by 20% trichloroacetic acid. Next, the collected–precipitated MP pellets were washed thrice with ethanol/ethyl acetate (1:1, v/v) solution. The absorbances of MPs dissolved in 6 M guanidine hydrochloride (pH 2.3) at 370 nm were read. The carbonyl content was calculated by the absorption coefficient of 22,000 M−1 cm−1 as nM/mg protein.”

2.6 Measurement of sulfhydryl content of MP

Delete the following paragraph, I think not needed as you already mentioned the reference of the published paper.

“Briefly, the protein samples, which were dissolved in urea–sodium dodecyl sulfate (SDS) solution (8.0 M urea, 3% SDS, 20 mM phosphate buffer, and pH 7.4), were incubated with DTNB reagent at room temperature for 15 min, followed by reading of the absorbance at 412 nm. The total sulfhydryl content was calculated by the absorption coefficient of 13,600M−1 cm−1 .”

2.7 Measurement of free-amine content

Delete the following paragraph, I think not needed as you already mentioned the reference of the published paper.

“A total of 200 mg OPA was dissolved in 128 mL 100 mM Na2B4O7 buffer (pH 9.75) containing 2 mL 100% ethanol. Then, 0.5 mL β-mercaptoethanol (β-ME) and 12.5 mL 10% SDS (w/v) were added to the solution, followed by diluting the volume to 250 mL with distilled water to prepare the OPA solution. Exactly 1 mL MP sample (2 mg/mL) was mixed with 6 mL OPA solution and reacted in the dark for 30 min.”

2.12 Surface hydrophobicity

Delete the following paragraph, I think not needed as you already mentioned the reference of the published paper.

In brief, the concentration of MP was adjusted to 5 mg/mL by suspending the MPs in 20 mM phosphate buffer with pH 6.0. Then, 1 mL resulting suspension and 0.2 mL of 1 mg/mL bromophenol blue (BPB) solution were mixed. The control was prepared in the same manner, but the MPs were replaced with 1 mL phosphate buffer. Before centrifugation (2000 g, 15 min, 4 °C), the solution was agitated for 10 min at 25 °C. The collected supernatant was diluted at a ratio of 1:10 by 20 mM phosphate buffer with pH 6.0.

Results & Discussion

I have read carefully the results and discussion section and I found that the appropriate discussion of the results is lacking.

P value should be italicized

Line 181-182: Delete the following sentence, not needed here

Carbonyl compounds are formed via oxidative deamination of susceptible side chains of amino acids, and carbonyl content is commonly used as a reliable indicator of protein oxidation [17]

Conclusion

This section is not adequate rewrite please

Author Response

Comments to the Author

Abstract

Followed by study objectives, please include material and methods (briefly) in this section prior to the results.

Reply: Thank you for your suggestions, we have added the material and methods briefly in abstract, as shown in 15-17. (This research aimed to explore the regularity of myofibrillar protein (MPs) oxidation mediated by MDA through fluorescence spectrum and thermogravimetric analysis, the binding capability and binding mode between MDA and MPs also investigated.)

Keywords: Keywords should be different from the title of the manuscript. Please replace the following keywords as they already appeared in the main title

“Malondialdehyde; Rabbit meat myofibrillar protein; Protein oxidation; Fluorescence quenching”

Reply: Thank you for your suggestions, we have replaced the keywords as they already appeared in the main title (Keywords: Mediated oxidation; Structural change; Addition reaction; Dynamic quenching), as shown in line 30.

Introduction:

The introduction section does not explain what was done on topic analyzed. Authors must focus that general information and improve this section by including the relevant information about the topic already available in the literature.

Line 34: The authors mentioned low fat content. What does it mean? Is it overall fat or Saturated fatty acids content? I recommend to include figures here showing a brief comparison of total fats with other species (beef, pork, chicken, mutton).

Line 35: Please verify this word “low allergenicity”

Is it correct? I think it should be low atherogenicity index?

Line 36: Why rabbit meat is more prone to microbial spoilage?

This point needs to be explained here with specific references.

Line 40: Please provide a suitable citation here.

Reply: Thank you for your suggestions, we have added the topic analyzed in introduction section. Our research is mainly focused on the protein oxidation induced by MDA, so the delete the low allergenicity and other extraneous description. The rabbit meat is more prone to microbial spoilage due to rich in protein. Besides, we have added the citation in line 40.

Materials & Methods:

This section is by far too long. Thus, detailed description of well-known/published methods is not necessary.

Please include a schematic diagram of the experimental design for better understanding of the readers. In the current form, it is hard to understand about the experimental design.

Line 59: Time of slaughtering? Because oxidation has a direct relationship with the time.

Line 62: Please specify the method of freezing.

Line 65: Please replace MP with MPs. Its plural, because its many proteins like actin, myosin, troponin, tropomyosin. Also replace was with were here.

Reply: Thank you for your suggestions, added schematic diagram of the experimental design, as shown in line 60-69 and 82-86, the samples were freeze in ultra-low temperature refrigerator, and the freezing LTL samples were thawed for 12 h in 4 °C incubator.

2.5 Measurement of carbonyl content of MP

Delete the following paragraph, I think not needed as you already mentioned the reference of the published paper that you are following this method without modification.

“Briefly, acidic conditions were attained to allow MP to react with DNPH. Then, the MP was precipitated by 20% trichloroacetic acid. Next, the collected–precipitated MP pellets were washed thrice with ethanol/ethyl acetate (1:1, v/v) solution. The absorbances of MPs dissolved in 6 M guanidine hydrochloride (pH 2.3) at 370 nm were read. The carbonyl content was calculated by the absorption coefficient of 22,000 M−1 cm−1 as nM/mg protein.”

2.6 Measurement of sulfhydryl content of MP

Delete the following paragraph, I think not needed as you already mentioned the reference of the published paper.

“Briefly, the protein samples, which were dissolved in urea–sodium dodecyl sulfate (SDS) solution (8.0 M urea, 3% SDS, 20 mM phosphate buffer, and pH 7.4), were incubated with DTNB reagent at room temperature for 15 min, followed by reading of the absorbance at 412 nm. The total sulfhydryl content was calculated by the absorption coefficient of 13,600M−1 cm−1 .”

2.7 Measurement of free-amine content

Delete the following paragraph, I think not needed as you already mentioned the reference of the published paper.

“A total of 200 mg OPA was dissolved in 128 mL 100 mM Na2B4O7 buffer (pH 9.75) containing 2 mL 100% ethanol. Then, 0.5 mL β-mercaptoethanol (β-ME) and 12.5 mL 10% SDS (w/v) were added to the solution, followed by diluting the volume to 250 mL with distilled water to prepare the OPA solution. Exactly 1 mL MP sample (2 mg/mL) was mixed with 6 mL OPA solution and reacted in the dark for 30 min.”

2.12 Surface hydrophobicity

Delete the following paragraph, I think not needed as you already mentioned the reference of the published paper.

In brief, the concentration of MP was adjusted to 5 mg/mL by suspending the MPs in 20 mM phosphate buffer with pH 6.0. Then, 1 mL resulting suspension and 0.2 mL of 1 mg/mL bromophenol blue (BPB) solution were mixed. The control was prepared in the same manner, but the MPs were replaced with 1 mL phosphate buffer. Before centrifugation (2000 g, 15 min, 4 °C), the solution was agitated for 10 min at 25 °C. The collected supernatant was diluted at a ratio of 1:10 by 20 mM phosphate buffer with pH 6.0.

Reply: Thank you for your suggestions, we have deleted the detailed description of the method.

Results & Discussion

I have read carefully the results and discussion section and I found that the appropriate discussion of the results is lacking.

P value should be italicized

Line 181-182: Delete the following sentence, not needed here

Carbonyl compounds are formed via oxidative deamination of susceptible side chains of amino acids, and carbonyl content is commonly used as a reliable indicator of protein oxidation [17]

Reply: Thank you for your suggestions, we have added the appropriate discussion of the results, as shown in line in line 163-166, 177-180, 245-251, 282-284 and 305-310.

The P has changed to italicized, and deleted the statement about carbonyl group, as shown in line 153 and 155.

Conclusion

This section is not adequate rewrite please

Reply: Thank you for your suggestions, we have revised the conclusion section, as shown in 361-367.

Round 2

Reviewer 3 Report

I am fully satisfied with the authors' response. 

Author Response

Thanks for the reviewer's comments